# *Staphylococcus aureus* Pneumonia: Preceding Influenza Infection Paves the Way for Low-Virulent Strains

**DOI:** 10.3390/toxins11120734

**Published:** 2019-12-17

**Authors:** Stefanie Deinhardt-Emmer, Karoline Frieda Haupt, Marina Garcia-Moreno, Jennifer Geraci, Christina Forstner, Mathias Pletz, Christina Ehrhardt, Bettina Löffler

**Affiliations:** 1Institute of Medical Microbiology, Jena University Hospital, Am Klinikum 1, D-07747 Jena, Germany; marina.garcia-moreno@med.uni-jena.de (M.G.-M.);; 2Section of Experimental Virology, Institute of Medical Microbiology, Jena University Hospital, Hans-Knöll-Str. 2, D-07745 Jena, Germany; Karoline.haupt@med.uni-jena.de (K.F.H.); christina.ehrhardt@med.uni-jena.de (C.E.); 3Institute for Infectious Diseases and Infection Control, Jena University Hospital, Am Klinikum 1, D-07747 Jena, Germany; christina.forstner@med.uni-jena.de (C.F.); mathias.pletz@med.uni-jena.de (M.P.); 4Department of Medicine I, Division of Infectious Diseases and Tropical Medicine, Medical University of Vienna, A-1090 Vienna, Austria; 5CAPNETZ Stiftung, D-30625 Hannover, Germany

**Keywords:** *Staphylococcus aureus*, pneumonia, influenza virus

## Abstract

*Staphylococcus aureus* is a facultative pathogenic bacterium that colonizes the nasopharyngeal area of healthy individuals, but can also induce severe infection, such as pneumonia. Pneumonia caused by mono- or superinfected *S.*
*aureus* leads to high mortality rates. To establish an infection, *S. aureus* disposes of a wide variety of virulence factors, which can vary between clinical isolates. Our study aimed to characterize pneumonia isolates for their virulent capacity. For this, we analyzed isolates from colonization, pneumonia due to *S. aureus*, and pneumonia due to *S. aureus*/influenza virus co-infection. A total of 70 strains were analyzed for their virulence genes and the host–pathogen interaction was analyzed through functional assays in cell culture systems. Strains from pneumonia due to *S. aureus* mono-infection showed enhanced invasion and cytotoxicity against professional phagocytes than colonizing and co-infecting strains. This corresponded to the high presence of cytotoxic components in pneumonia strains. By contrast, strains obtained from co-infection did not exhibit these virulence characteristics and resembled strains from colonization, although they caused the highest mortality rate in patients. Taken together, our results underline the requirement of invasion and toxins to cause pneumonia due to *S. aureus* mono-infection, whereas in co-infection even low-virulent strains can severely aggravate pneumonia.

## 1. Introduction

*Staphylococcus aureus* (*S. aureus*) is a facultative pathogenic bacterium that colonizes the nasopharyngeal area of many healthy individuals (about 30% of the population [1,2]), but can also cause severe infections, such as pneumonia and bloodstream infections aggravating to sepsis and septic shock [3]. Community-acquired pneumonia (CAP) is the most frequent infection requiring hospitalization and accounts for high morbidity and mortality worldwide. In children, CAP is the leading cause of death [4], and also in the elderly (over 80 years of age) the incidence of pneumonia is high [5]. The problem is further aggravated by the emergence and spread of methicillin-resistant *S. aureus* (MRSA). CAP caused by *S. aureus* is relatively rare compared to pneumococcal CAP [6] but often very severe. In contrast to CAP, in hospital-acquired pneumonia (HAP) *S. aureus* is the most frequent pathogen.

While CAP and HAP were traditionally considered to be bacterial infections, novel diagnostic tools uncover a hidden burden of viral pathogens [7]. Viral pathogens, such as rhinovirus, respiratory syncytial virus, and influenza virus (IV) are common causes of pneumonia, whereas IV is mostly dreaded, as influenza repeatedly causes big pandemics with high mortality worldwide [8]. During the last influenza pandemics, in particular, critically ill influenza patients were often superinfected by bacterial pathogens (up to 20% of all influenza patients) [9,10], with *S. aureus* being very common (about 40%) [11]. Bacterial superinfection is often associated with severe illness and acute respiratory distress syndrome [11]. In some cases, even necrotizing pneumonia with a high mortality rate was reported [12,13].

*S. aureus* is a very versatile pathogen that can cause a wide array of infections due to its multitude of virulence factors [3]. The genes for virulence factors and their expression can largely vary between clinical isolates. In the pathogenesis of pneumonia, not a single *S. aureus* virulence factor could be identified as being causative. However, some bacterial toxins were reported to contribute to lung infections, such as the pore-forming α-hemolysin (Hla) [14], the leucocidins (e.g., Panton–Valentine leukocidin (PVL)) [15], and the phenol-soluble modulins (PSMs) [16]. Toxins are mainly associated with inflammation, cell death induction, and tissue destruction that can explain the clinical picture of acute pneumonia. However, as most of the *S. aureus* isolates express multiple toxins, which act together in a complex way [17], tissue destruction is most likely the result of many toxic mechanisms [18]. Another group of virulence factors that contribute to infection development is the class of adhesins, e.g., the fibronectin-binding proteins (FnBPs) [19]. Adhesins are staphylococcal surface components that provide tight adherence of the bacteria to the extracellular matrix and host cells. Adherence to host cells can be followed by bacterial uptake. In the last decade, *S. aureus* has been increasingly recognized as an intracellular pathogen, which contributes to many types of tissue infection [20].

Many types of virulence factors have been demonstrated to account for the development of lung infections. However, most of these data were obtained from in vitro and in vivo infection models [15,21], which demonstrate the impact of a defined virulence factor but do not reveal which virulence strategies dominate in patients. This study aimed to analyze *S. aureus* clinical isolates from (i) colonization of healthy volunteers, (ii) patients with pneumonia (CAP or HAP) due to *S. aureus* mono-infections, and patients with pneumonia due to an *S. aureus* and IV co-infection. We analyzed the strains genotypically for the presence virulence factors and tested the strains in functional phenotypic assays for host cell invasion, biofilm formation, and cytotoxicity. We found that strains from pneumonia reveal a higher invasive capacity and higher cytotoxicity against immune cells than strains from colonization and co-infection, suggesting that bacteria require a defined level of virulence to initiate a lung infection. By contrast, *S. aureus* strains being involved in co-infection with IV need a much lower level of virulence for superinfection, comparable to that of colonizing strains. Our study points to the role of toxins and adhesins in pneumonia development, but also indicates that a preceding viral infection is a dangerous situation, as it can pave the way for much less virulent (only colonizing) *S. aureus* strains to aggravate pneumonia.

## 2. Results

### 2.1. The Genetic Analysis of Staphylococcal Isolates Showed That Primary Pneumonia Is Associated with Certain Exotoxins and Proteases

For the genetic analysis, 70 *S. aureus* isolates were selected, 20 from nasopharyngeal colonization (colonization), 20 from CAP-patients through the CAPNETZ study [22], 20 from HAP of patients from the Jena university hospital (CAP, HAP = primary pneumonia), and 10 from IAV (influenza A virus) pneumonia with an *S. aureus* superinfection (co-infection). The main clinical characteristics of all patients are summarized in Table 1. All strains were analyzed with the PanStaph Genotyping microarrays (Alere Technologies GmbH, Jena, Germany), facilitating the detection of specific genes, as well as their assignment to clonal complexes (CCs).

It was remarkable that the isolates from colonization and co-infection contained fewer CCs than the strains isolated from pneumonia (CAP and HAP) (Figure 1A–D). The distributions in both pneumonia groups were very diverse, as they contained at least six different CCs. The determination of the clonal complexes of the isolated *S. aureus* strains shows different distributions patterns between the colonization/co-infection strains and the strains isolated from pneumonia. Within all investigated groups we can detect the following CCs: CC8, CC15, CC30, CC45. Exclusively isolated from *S. aureus* pneumonia were the following complexes: CC7, CC12, CC121, CC96.

The complete microarray hybridization data are sorted in main groups of virulence factors (e.g., adhesins, toxins) and are demonstrated in Table 2. The presence of certain toxin genes was significant in pneumonia isolates, as outlined below. Some superantigens and proteases were more common in isolates from pneumonia due to *S. aureus* mono-infection than in isolates from colonization or co-infection., suggesting that these virulence factors could play an important role in pneumonia development.

Many genes encoding the capsule (e.g., *cap 5, 8*), biofilm (*ica A*, *C*, *D*), and adhesins were present in all isolates tested. Interestingly, the surface protein involved in biofilm formation and capsule type 1 was not detected within all groups. Only some genes (e.g., *fnbpB*) were not present in all isolates but were distributed equally among the four groups of strains.

Taken together, our results show that similar patterns of genes can be found in strains from colonization and co-infection, whereas strains from pneumonia due to a mono-infection are characterized by genes for defined toxins and exoproteins.

### 2.2. Strains Obtained From Primary Pneumonia Are More Invasive in Host Cells Than Strains from Colonization and Co-Infection

As the presence of genes does not provide any information on the expression and function of defined virulence factors, we tested in a next step selected strains for their activity in the host–pathogen interaction. For this, we performed functional assays in cell culture systems. Initially, we focused on pathogen characteristics that are important to initiate an infection. *S. aureus* can adhere to host structures including host cells, which is followed by the uptake of the bacteria within the intracellular location [23]. Recent studies have demonstrated that host cell invasion essentially contributes to *S. aureus* virulence [20]. We measured host cell invasion of selected isolates, 10 from colonization, 10 from CAP, 10 from HAP, and 10 from co-infection with IV, in primary endothelial cells (human umbilical vein endothelial cells (HUVEC)). We found that strains from *S. aureus* mono-infection showed higher host cell invasion than strains from colonization and co-infection (Figure 2A).

Host cell invasion is mainly mediated by the adhesins FnBPs (*fnbpA and fnbpB)* [24], which are present in almost all *S. aureus* isolates (Table 1). However, the adhesins are also involved in further pathogenic strategies, such as biofilm formation [25]. As a second functional assay, we tested the ability of the different strains to form a biofilm by an in vitro assay. We were able to detect a higher capacity for biofilm formation for the mono-infected *S. aureus* strains (HAP) compared to the colonization and co-infection isolates (Figure 2B).

Overall, we demonstrate that host cell invasion is a characteristic feature of strains that establish pneumonia.

### 2.3. Strains Obtained from Primary Pneumonia Exhibit Higher Cytotoxicity against Immune Cells Than Strains from Colonization and Co-Infection

Finally, we tested the cytotoxic capacity of the different isolates against non-professional and professional phagocytes. As non-professional phagocytes, we isolated human umbilical vein endothelial cells (HUVEC) freshly from umbilical cords to use them as a model system for primary isolated cells. We stimulated confluent cell cultures with live bacteria of the different strains. After 1 h we removed all extracellular bacteria by lysostaphin-treatment and incubated the infected host cells for 1 d. After washing, we detached the endothelial cells and measured cell death induction by flow cytometry analysis. Comparing all strains from the four groups we did not detect any differences in the ability of the *S. aureus* isolates to induce cell death in endothelial cells (Figure 3A).

As professional phagocytes, we isolated neutrophils from different healthy volunteers and stimulated the cells with the supernatants of the different strains. After 1 h of exposure, we measured cell death by FACS analysis. We found that supernatants from strains obtained from CAP and HAP pneumonia strains induced significantly higher cell death in neutrophils than supernatants from colonization or co-infection strains (Figure 3B).

Taken together, the cytotoxic effects of pneumonia isolates against immune cells can be correlated with the high presence of genes for exoproteins and toxins (Table 1) that are secreted and are mainly directed against professional phagocytes but do not act against non-professional phagocytes, such as endothelial cells.

## 3. Discussion

Pneumonia caused by *S. aureus* is a significant cause of morbidity and mortality and can induce severe lung destruction [4]. *S. aureus* is endowed with a myriad of virulence factors that enable the bacteria to adhere to host structures, escape from the host immune system, invade host cells, and protect themselves from clearance [3]. In all types of invasive tissue infections, different virulence factors act in concert to establish and maintain an infection. In *S. aureus* pneumonia, a crucial and essential virulence factor was never defined that could be addressed as a therapeutic target or by vaccination [26]. Consequently, the aim of our study was to identify pathogen strategies mediated by the combined action of several bacterial factors that are important in pneumonia development. We found that *S. aureus* isolates causing pneumonia in a mono-infection were characterized by (i) high host cell invasion and (ii) cytotoxicity against immune cells.

In the last decades, *S. aureus* has increasingly been recognized as an intracellular pathogen [23,27]. The main adhesins that account for host cell invasion are the FnBPs that are widely expressed among *S. aureus* isolates [28]. However, many alternative adhesins support the internalization process [29] and participate in biofilm formation. For most *S. aureus* strains obtained from infection and colonization, high invasive capacity in different host cell types was reported [30,31]. In our present study, we even detected a difference between colonizing and pneumonia strains that point to the importance of host cell invasion in lung infection. Within the lung, bacteria could use the intracellular location to hide from the host immune cells and to induce host cell destruction to spread. In addition, biofilm formation contributes to the protection against the immune response.

Our genetic analysis and cytotoxic assays showed that *S. aureus* strains obtained from CAP and HAP hold a high capacity to defend against immune cells. In these pneumonia isolates, the toxin-genes for the superantigens 2, 3, 9 were significantly more present than in colonizing strains. These toxins are preferentially directed against immune cells to induce inflammation and cell death. With respect to superantigens, different models have demonstrated enhanced activation and cytotoxicity against the immune system that contributes to lung infection [32,33]. The efficient bacterial defense against immune cells is of particular importance during lung infection, as immune cells are abundantly recruited to lung tissue during infection and the interaction of *S. aureus* with the immune system appears to be a key factor in the pathogenesis of pneumonia [34]. Furthermore, some secreted protein/exoenzymes were found highly prevalent in pneumonia strains, such as the *serine protease A* and *B*. These exoenzymes have been demonstrated to promote bacterial spread in a rabbit model of pneumonia [35]. Additionally, the zinc metalloproteinase *aureolysin* is significantly more prevalent within the pneumonia strains. As a complement inhibitor, aureolysin is associated with immune evasion [36,37].

More than 100 CCs of *S. aureus* are known. The most frequent clusters associated with hospital-acquired infections were CC5, CC8, CC22, CC30, and CC45 [38]. In general, the clonal population structure of our isolates was comparable to findings reported in other studies [35]. Our results show, that the most prevalent clusters of our study were CC30 and CC45. In addition, CC8 could be also detected within all four investigated groups. CC8 is the clonal structure of highly prevalent strains of the United States, such as USA300, and it is the most prevalent strain in health care institutions [39]. Furthermore, we detected CC121 only in the CAP and HAP strains. CC121 is highly prevalent and it is associated with superficial skin infections but also severe invasive infections [40].

A limitation of our study is the limited number of strains that were able to be analyzed in depth with the labor-intensive functional assays. We could, however, demonstrate that the presence of defined virulence genes is associated with pneumonia due to mono-infection in patients and that these characteristics can be reproduced by functional assays. Due to our small sample size, we cannot exclude that other virulence factors are also involved in lung infection, but our study shows a clear tendency that toxins directed against immune cells and exoproteases contribute to settling an *S. aureus* infection in lung tissue. As clinical approaches failed to address a single virulence factor, e.g., by vaccine development, whole-pathogen strategies that are mediated by the concert action of several virulence factors should be in the focus of novel therapeutic approaches.

Another aspect that is outlined in our study is viral–bacterial co-infection. A preceding IV- infection is known to increase susceptibility to secondary bacterial pneumonia [10,11,41]. In the recent influenza epidemics, *S. aureus* has been reported as the most common bacterial superinfecting pathogen, which can result in severe disease courses with often fatal outcome [42]. For this study, we collected *S. aureus* strains of IV/*S. aureus* co-infection of the Jena University Hospital during the 2017/2018 influenza season and reached the number of 10 strains. We analyzed these strains in parallel to mono-infecting pneumonia strains and colonizing strains and obtained striking results. In general, the co-infecting strains resembled the strains obtained from the nasal colonization of healthy volunteers and they did not reveal the characteristics of pneumonia strains (CAP and HAP), such as high invasion and increased cytotoxicity against immune cells. This phenomenon can be explained that during a co-infection the bacteria start to settle in a pre-damaged lung. Bacteria can take advantage of exposed adherence sites, of a compromised immune system, and of a nutrient-rich environment [11]. Consequently, superinfecting *S. aureus* strains do not require to defend against an active immune system and to hide from immune function; which means that every colonizing *S. aureus* strain can induce a bacterial superinfection. Our results clearly show that the mortality rates of co-infection are dramatically increased compared to that of the single infection.

Taken together, our study based on clinical *S. aureus* isolates suggests that invasive, proteolytic, and cytotoxic virulence factors against immune cells are required to establish a lung infection. If the host is impaired due to a preceding IV-infection, much less virulence is required for bacterial superinfection. Our data suggest that a primary virus infection facilitates a secondary bacterial infection for even low-virulent strains. Our study underlines the weak and susceptible patients’ state during influenza, as any colonizing *S. aureus* strain can cause a severe superinfection.

## 4. Materials and Methods

### 4.1. Study Design and Informed Consent

The analyzed S. aureus strains were obtained from a cross-section of healthy volunteers and patients suffering from pneumonia. We collected strains of nasopharyngeal colonization ordered by age (mean age 61.80 (± 19.32)) years from nasopharyngeal swabs of healthy volunteers from the region of Jena city. For the pneumonia strains, we collected isolates obtained by the CAPNETZ network (mean age 64.40 (± 9.58) years, German patients treated on an ambulatory basis, German Clinical Trials Register: DRKS00005274) [43] and from hospitalized patients from the Jena University Hospital (mean age 62.40 (± 6.34) years). The S. aureus strains co-infecting a primary influenza A virus (IAV) infection were collected within the 2016/2017 season within the Jena University Hospital. For this, patients with a positive bronchoalveolar lavage (BAL) were recruited and S. aureus was isolated and stored under standard conditions at −80 °C. A total of 70 persons were included in this cross-sectional study; all healthy probands did not show signs of clinical symptoms of infection at the time of taking swabs. The study was approved by the ethics committee of the Jena University Hospital, Germany (no.: 4449-06/15, 4765-04/16). All samples were obtained from patients over 18 years. The CAPNETZ protocol was approved by the ethical review boards of each participating clinical center (approval number of leading ethics committee Medical Faculty of Otto-von-Guericke-Universität Magdeburg, no. 104/01, see Acknowledgments or www.capnetz.de for participating centers). Informed consent was obtained from all individuals participating in the study in accordance with the Declaration of Helsinki.

### 4.2. Sample Preparation

For the cultivation of the nasopharyngeal swabs, S. aureus was plated on Columbia agar plates (5% sheep blood) and incubated at 37 °C for 24 h. The identification and resistance determination were carried out by using matrix-assisted laser desorption/ionization (MALDI), and time-of-flight (TOF) analyzer (MALDI-TOF) and VITEK MS (bioMérieux, Marcy l ‘Ètoile). For each group, 20 S. aureus isolates of colonization, CAP, and HAP-strains and 10 co-infection strains were randomly selected and subjected to genotyping. By using genotyping, we can exclude clonal identity of the examined strains. For phenotypic analysis, we selected 10 strains from each group. All S. aureus strains from co-infection were isolated from IAV-infected patients, detected with qRT-PCR (QIAGEN, Hilden, Germany).

### 4.3. Functional Cell-Based Assays

To determine the phenotypic characterization of the individual virulence of the isolated strains, we performed functional cell culture-based assays.

First, the rate of cell death was determined by analyzing the proportion of apoptotic/necrotic cells with hypodiploid nuclei (apoptotic nuclei after propidium iodide staining (BD, Heidelberg, Germany)) [14]. For this, human endothelial cells isolated from umbilical cords (HUVEC) and polymorphonuclear neutrophils (PMNs) were used. For the stimulation of the professional cells, freshly isolated PMNs were sown at 1 × 10^6^ cells per neutrophil and incubated for 60 min with 30% of the bacterial supernatants, 1% serum, and 1 mM (4-(2-hydroxyethyl)-1-piperazineethanesulfonic acid) (HEPES) at 37 °C and 5% CO_2_. Thereafter, staining with propidium iodide (Becton Dickinson, Franklin Lakes, NJ, USA) for assessment of cell viability by flow cytometry was performed.

The analyzation of the internalization process was carried out by using HUVEC [14]. After infection of the cell cultures, the invasion was measured by flow cytometry, whereby the invasiveness of the laboratory strain Cowan I was set as 100% [15].

The biofilm formation was measured after 48 h of incubation in TBS + 0.25% glucose medium. Before the measurement, they were washed and stained with crystal violet 1% and the crystal violet was dissolved with ethanol/acetone (80%/20%). Afterward, the absorbance was measured at 570 nm with an ELISA TECAN reader. The laboratory strains TM300 and RP62A were selected as negative and positive controls for biofilm formation respectively.

### 4.4. Genotyping

For the detection of S. aureus genes, we use the PanStaph Alere Genotyping Kit 2.0 (Alere Technologies GmbH, Jena, Germany) as described previously [1]. Here, the simultaneous detection of 330 genes (resistance, virulence and housekeeping genes) including a cluster analysis was done.

### 4.5. Statistical Analysis

Statistical analysis was performed with GraphPad Prism software version 7 and 8 (GraphPad Software, La Jolla, CA, USA) or Microsoft Excel (Microsoft Office, 2010). A comparison between more than two groups was performed using a one-way or two-way ANOVA test. Where indicated, a comparison between two groups was performed using two-way ANOVA with Tukey’s multiple comparisons test.

## Figures and Tables

**Figure 1 toxins-11-00734-f001:**
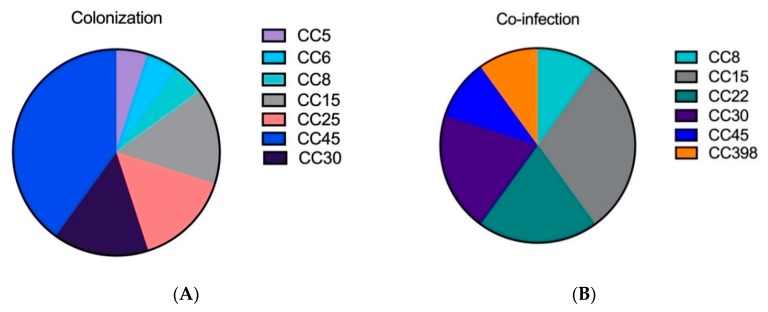
Clonal complexes (CC) of *Staphylococcus aureus* strains isolated from (**A**) colonization, (**B**) co-infection with influenza virus, (**C**) CAP—community-acquired pneumonia, and (**D**) HAP—hospital-acquired pneumonia determined with the PanStaph Genotyping Kit (Alere Technologies GmbH, Jena, Germany).

**Figure 2 toxins-11-00734-f002:**
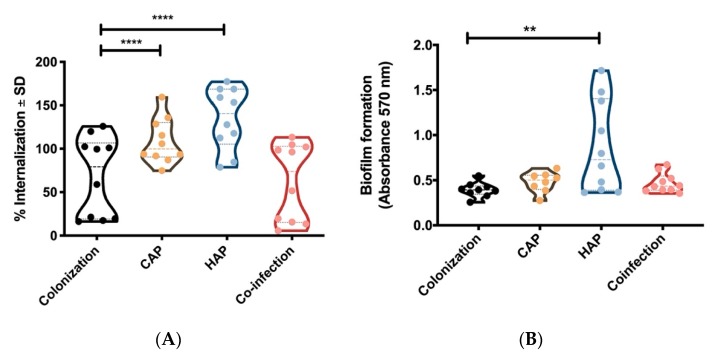
(**A**) Internalization ability (in %) and (**B**) biofilm formation of each of 10 *S. aureus* strains isolated from nasopharyngeal swabs (colonization), CAP, HAP, and from co-infection with IV. Two-way ANOVA with Tukey’s multiple comparisons test ** *p* < 0.01, **** *p* < 0.000.

**Figure 3 toxins-11-00734-f003:**
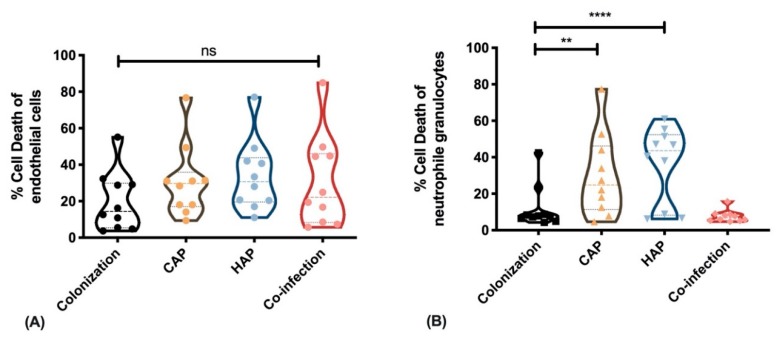
(**A**) Cell death (in %) carried out on endothelial cells and (**B**) on neutrophil granulocytes induced with 10 *S. aureus* strains each isolated from nasopharyngeal swabs (colonization), CAP, HAP, and from co-infection with IV. Two-way ANOVA with Tukey’s multiple comparisons test: ns = non-significant, ** *p* < 0.01, **** *p* < 0.0001.

**Table 1 toxins-11-00734-t001:** Characteristics of 20 healthy volunteers (colonization) and 50 patients (CAP, HAP, co-infection with influenza A virus) from whom the bacteria were isolated.

Characterization	Colonization	CAP	HAP	Co-Infection
Age (mean ± SD)	61.80 (±19.32) years	64.40 (±9.58) years	62.40 (±6.34) years	58.10 (±6.99) years
Male Sex	60%	70%	55%	40%
Specimen	100% respiratory specimens	50% respiratory specimens 50% BAL	15% respiratory specimens 85% BAL	10% respiratory specimens 90% BAL
Mortality	0%	20%	15%	60%

**Table 2 toxins-11-00734-t002:** Overview of *S. aureus* virulence genes from isolates of colonized volunteers and patients with CAP, HAP, and co-infection with influenza virus determined by PanStaph Genotyping Kit (Alere Technologies GmbH, Jena, Germany). Chi-square test, strains with significant more detectable genes compared between colonization strains and CAP/HAP are marked (0.0332 (*); 0.0021 (**)).

Virulence Factors				% Positive Isolates		
Group	Gene	Description	Colonization (n = 20)	CAP (n = 20)	HAP (n = 20)	Co-Infection (n = 10)
**Hemolysin gamma and Leukocidins**	*Tst1*	Toxic Shock Toxin 1	10	25	15	20
	*lukX*	Leukocidin/Hemolysin Toxin Family Protein X	100	100	100	90
	*lukY*	Leukocidin/Hemolysin Toxin Family Protein Y	100	100	100	100
	*lukE*, *lukD*	Leukocidin E/D	45	65	70	30
	*lukF*, *lukS*	Hemolysin gamma (comp. B, C)	100	100	100	100
	*lukF/S-PV*	Panton Valentine Leukocidin	5	0	0	0
**Hemolysin**	*hla*	Hemolysin alpha	100	95	100	100
	*hlb*	Hemolysin beta	65	80	90	60
	*hlIII*	Putative membrane proteins	100	100	100	100
**Staphylococcal Superantigen**	*ssl02*	Staphylococcal Superantigen-like Protein 2	50	65	80 *	40
	*ssl03*	Staphylococcal Superantigen-like Protein 3	45	70	90 **	30
	*ssl07*	Staphylococcal Superantigen-like Protein 7	45	60	60	30
	*ssl08*	Staphylococcal Superantigen-like Protein 8	40	70	70	30
	*ssl09*	Staphylococcal Superantigen-like Protein 9	50	65	80 *	40
	*ssl11*	Staphylococcal Superantigen-like Protein 11	55	15	20	0
***Hlb*-modificated phages and proteases**	*chp*	Chemotaxis-inhibiting protein	75	60	55	80
	*aur*	Aureolysin	60	90 *	100 **	100
	*splA*	Serinprotease A	45	65	70	30
	*splB*	Serinprotease B	45	70	70	30
	*sspB/A/P*	Staphopain B/A/P, Protease	100	100	100	100
**Adhesion factors**	*fnbpA*	Fibronectin-binding protein A	100	100	100	100
	*fnbpB*	Fibronectin-binding protein B	85	70	80	50
	*clfA*	Clumping factor A	100	100	100	100
	*clfB*	Clumping factor B	100	100	100	100
	*cna*	Collagen binding adhesin	65	60	50	60
	*ebh*	Cell wall associated fibronectin-binding protein	100	90	95	70
	*ebpS*	Cell surface elastin binding protein	100	100	100	100
	*eno*	enolase	100	100	100	100
	*sasG*	*S. aureus* surface protein G	30	45	45	50
	*vwb*	Von Willebrand factor binding protein	100	100	100	100
	*eap*	Extracellular adherence protein	100	95	100	100
	*sdrC*	Ser-Asp rich fibrinogen-/bone sialoprotein-binding protein C	100	100	100	100
	*sdrD*	Ser-Asp rich fibrinogen-/bone sialoprotein-binding protein D	70	90	90	80
**Capsule/Biofilm**	*Cap 1*	Capsule type 1	0	0	0	0
	*Cap 5*	Capsule type 5	30	30	40	40
	*Cap 8*	Capsule type 8	75	70	60	60
	*ica A*	intercellular adhesion protein A	100	100	100	100
	*ica C*	intercellular adhesion protein C	100	100	100	100
	*ica D*	biofilm polysaccharide intercellular adhesin (PIA) synthesis protein D	100	100	100	100
	*bap*	surface protein involved in biofilm formation	0	0	0	0
	*bbp*	bone sialoprotein-binding protein	85	90	100	90

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
