# Peer review of "Staphylococcus aureus Pneumonia: Preceding Influenza Infection Paves the Way for Low-Virulent Strains"

_toxins, 2019, doi:10.3390/toxins11120734_

Round 1
Reviewer 1 Report
Summary – The manuscript submitted characterizes 70 clinical S. aureus strains for the presence of virulence factors depending on source. 20 nasal, 20 HAP, 20 CAP, and 10 influenza co-infection isolates were tested. Pneumonia isolates were dissimilar with nasal colonization or co-infection isolates in terms of some virulence genes. Host cell invasion, biofilm formation, and the ability to induce PMN death were also elevated in pneumonia isolates. These data suggest that monoinfection with S. aureus is characterized by differential virulence factor expression from colonization or co-infection. Also, co-infection strains are highly pathogenic despite lower virulence factor expression than pneumonia strains.
Major Comments –
1) Study Design – The rationale for performing cell attachment studies in HUVEC cells is lacking. An airway epithelial cell line such as HBE would be more appropriate.
2) Interpretation of Genetic Data – Relatively few genes are different between the isolates and as a primary finding, the conclusions are somewhat overstated. Functional data are more compelling. Measurement of virulence factor gene expression or protein in the in vitro assays would greatly improve the conclusions being made.
3) Pathoadaptation – The authors conclude that preceding virus paves the way for low virulence S. aureus, however is it not possible that S. aureus in co-infection has selected against virulence genes as a means of pathoadaptation. This should be discussed.
Minor Comments –
1) Figure 2 – This figure is mislabeled as figure 3 in the figure legend and text.
2) Figure 3 – This figure is mislabeled as figure 4 in the text.
Reviewer 2 Report
Minor points;
Your abstract should have a little more detail on type of patient (hospital, ICU, community) and age bracket (adult, pediatric) and also the range of virulence assays.
Also time of year – influenza season? Influenza A? non-A?
Line 26 “.. for colonizing S. aureus strains to induce a super-infection.” Suggest “….for colonizing S. aureus strains to induce an opportunistic infection.”
Figure 1 is difficult to compare the infection types. Suggest four histograms stacked on top of each other.
Figure 3 (also figure 4 mislabelled as figure 3) is presumably a violin plot – this needs a little more description.
Line 209/10 “A limitation of our studs is the limited number of strains that were therefore analyzed with labor intensive functional assays in depth.”suggest “A limitation of our study is the limited number of strains that were able to be analyzed in depth with the labor intensive functional assays.”
Round 2
Reviewer 1 Report
The authors have addressed my concerns.